# Zoledronic Acid Blocks Overactive Kir6.1/SUR2-Dependent K_ATP_ Channels in Skeletal Muscle and Osteoblasts in a Murine Model of Cantú Syndrome

**DOI:** 10.3390/cells12060928

**Published:** 2023-03-17

**Authors:** Rosa Scala, Fatima Maqoud, Conor McClenaghan, Theresa M. Harter, Maria Grazia Perrone, Antonio Scilimati, Colin G. Nichols, Domenico Tricarico

**Affiliations:** 1Sections of Pharmacology, Medicinal Chemistry, Department of Pharmacy—Pharmaceutical Sciences, University of Bari “Aldo Moro”, 70125 Bari, Italy; 2Center for the Investigation of Membrane Excitability Diseases, Department of Cell Biology and Physiology, Washington University School of Medicine, St. Louis, MO 63110-1010, USA

**Keywords:** Cantú syndrome, rare disease, ATP-sensitive potassium channel, skeletal muscle, glibenclamide, patch clamp, zoledronic acid, anti-cancer drug

## Abstract

Cantú syndrome (CS) is caused by the gain of function mutations in the *ABCC9* and *KCNJ8* genes encoding, respectively, for the sulfonylureas receptor type 2 (SUR2) and the inwardly rectifier potassium channel 6.1 (Kir6.1) of the ATP-sensitive potassium (KATP) channels. CS is a multi-organ condition with a cardiovascular phenotype, neuromuscular symptoms, and skeletal malformations. Glibenclamide has been proposed for use in CS, but even in animals, the drug is incompletely effective against severe mutations, including the Kir6.1^wt/V65M^. Patch-clamp experiments showed that zoledronic acid (ZOL) fully reduced the whole-cell KATP currents in bone calvaria cells from wild type (WT/WT) and heterozygous Kir6.1^wt/V65M^CS mice, with IC_50_ for ZOL block < 1 nM in each case. ZOL fully reduced KATP current in excised patches in skeletal muscle fibers in WT/WT and CS mice, with IC_50_ of 100 nM in each case. Interestingly, KATP currents in the bone of heterozygous SUR2^wt/A478V^ mice were less sensitive to ZOL inhibition, showing an IC_50_ of ~500 nM and a slope of ~0.3. In homozygous SUR2^A478V/A478V^ cells, ZOL failed to fully inhibit the KATP currents, causing only ~35% inhibition at 100 μM, but was responsive to glibenclamide. ZOL reduced the KATP currents in Kir6.1^wt/VM^CS mice in both skeletal muscle and bone cells but was not effective in the SUR2^[A478V]^ mice fibers. These data indicate a subunit specificity of ZOL action that is important for appropriate CS therapies.

## 1. Introduction

Cantú Syndrome (CS) is a rare autosomal-dominant, a multi-organ condition characterized by cardiomegaly, vascular dilation, low blood pressure, hypertrichosis, neuromuscular symptoms, and skeletal malformations [1,2,3]. CS is caused by gain-of-function (GOF) mutations in the *ABCC9* and *KCNJ8* genes [4,5], encoding the SUR2 and Kir6.1 subunits, respectively, of ATP-sensitive potassium (KATP) channels. To date, >70 genetically confirmed individuals have been reported with CS and associated with >30 missense *ABCC9* or *KCNJ8* mutations [6,7], yet there is currently no recommended therapy for CS. Sulfonylurea inhibitors are obvious candidate therapies. Glibenclamide concentrations dependently reduce the KATP channel current of wild-type (WT) and CS mutant channels in heterologous expression systems, as well as in cardiac, smooth, and skeletal muscles cells of CS mutant mice in which the SUR2^A478V^ mutation is engineered into the mouse locus using CRISPR/Cas9 [8,9]. The reversal of cardiovascular phenotypes in SUR2^A478V^ mice [10] and the apparent benefit of glibenclamide, with minimal disruption of glycemic control in one human case [10] are consistent with this action. However, the inhibitory effect of glibenclamide is reduced in mutant channels with more severe GOF, particularly in Kir6.1 GOF mutants [11,12]. Accordingly, glibenclamide is much less effective against KATP channels in cells from CS mutant mice carrying the Kir6.1^V65M^ mutation [8,9,12]. These data suggest that while high doses of glibenclamide, with careful monitoring of glycemia, may be effective in patients carrying *ABCC9* mutations, *KCNJ8* variant CS patients may not benefit.

We recently reported that in SUR2^wt/A478V^ and SUR2^A478V/A478V^ mice, forelimb strength was reduced, and this was associated with metabolic decoupling and atrophy in different muscles [12]. A slight rightward shift of sensitivity to inhibition by glibenclamide was detected in SUR2^A478V/A478V^ mice, which is indicative of a conserved response to this drug. However, in Kir6.1^V65M^ mice, skeletal muscle was severely affected, muscles were atrophic and necrotic, and there was marked loss of muscle mass with fibrotic replacement, inflammatory cell infiltration, up-regulation of autophagy genes, and reduced muscle strength in slow twitch muscle, as well as severe impairment of glibenclamide response- [9]. These symptoms are very different from what previously observed in other neuromuscular disorders known as Hypokalemic Periodic Paralysis associated with the loss of function of the fast-twitch KATP channels in humans [11,13] and animals [14,15]. The human patients and the animals show muscle weakness with vacuolar features [16], but no necrosis or inflammation, and flaccid paralysis induced by insulin injection is associated with the lowering of extracellular K^+^ ions concentration [17,18]. The phenotype is reversed by the use of KATP channel openers and calcium-activated potassium channel openers [16,19,20] used in cardiovascular diseases [20], proposed for use in analgesia, and used to increase drug delivery in the brain in cancer [21,22].

CS patients also exhibit a large variety of bone malformations [6], suggesting that KATP channels may be important in bone development. Few reports are available in the literature regarding KATP channel expression in bone cells, and there are no data on their function. We recently showed that zoledronic acid (ZOL) is a potent blocker of KATP channels in bone and skeletal muscle cells in excised patch experiments and whole-cell patches, respectively [23]. Experimentally, the drug showed selectivity against the muscular and osteoblast KATP channels vs. the pancreatic subtype. In silico investigation suggests that the action of ZOL is mediated by interactions with multiple sites on the KATP complex, including the ADP and ATP binding sites on SUR2 and Kir6.1 subunits and the sulfonylureas binding site on SUR2 [23,24].

Here, we investigated the action of zoledronic acid, a musculoskeletal drug, targeting KATP channels of muscle fibers and bone cells of Kir6.1^V65M^ and SUR2^A478V^ CS mice to evaluate the possible use in the treatment of the musculoskeletal symptoms in CS.

## 2. Materials and Methods

### 2.1. Animal Care

As previously described, CRISPR/Cas9 gene editing was used to introduce single-nucleotide mutations into the endogenous *ABCC9* and *KCNJ8* gene loci, resulting in protein substitutions that are analogous to SUR2^wt/A478V^ (A476V in mouse sequence) and Kir6.1^V65M^ in human CS patients [8]. Heterozygous (SUR2^wt/A478V^) and homozygous (SUR2^A478V/A478V^) SUR2^A478V^ mice, as well as heterozygous Kir6.1^V65M^ (Kir6.1^wt/V65^) mice, were generated as previously described [8]. Pathogen-free mice were imported from Washington University in Saint Louis, under approved protocols at the Stabulario of the Dipartimento di Farmacia-Scienze del Farmaco, University of Bari, Bari, Italy, under the supervision of the veterinary officer under D.lgs. 26/2014. Mice were maintained 2–4 per cage and provided a standard laboratory diet and water ad libitum. The laboratory was kept at 50 ± 5% relative humidity and at a temperature of 22 ± 1 °C under 12:12 light/dark cycles.

### 2.2. Ethical Statements

All the applied experimental protocols and animal care were in compliance with the European Directive 2010/63/EU on Animal Protection Used for Scientific Experiments and the Washington University School of Medicine Institutional Animal Care and Use Committee. They were approved by the Italian Ministry of Health and by the Committee of the University of Bari O.P.B.A (Organization for Animal Health) (prot. 8515-X/10, 30 January 2019).

### 2.3. Drugs and Solutions

The KATP channels blocker glyburide/glibenclamide cat. N° PHR1287 and BaCl_2_ cat. N° 449644 were purchased from Sigma (SIGMA Chemical Co., Milan, Italy) as well as all chemicals. The zoledronic acid (ZOL) was synthesized, purified in our labs, and dissolved in phosphate-buffered saline (PBS) stock solution at 1 or 10 × 10^−3^ M concentrations.

The normal Ringer solution used during muscle and organ biopsy contained 145 mM NaCl, 5 mM KCl, 1 mM MgCl_2_, 0.5 mM CaCl_2_, 5 mM glucose, and 10 mM 3-(N-morpholino) propane sulfonate (Mops) sodium salt and was adjusted to pH 7.2 with Mops acid. The solutions for excised patch experiments on isolated muscle fibers were as follows. The patch-pipette solution contained 150 mM KCl, 2 mM CaCl_2_, and 1 mM Mops (pH 7.2); the bath solution contained 150 mM KCl, 5 mM EGTA, and 10 mM Mops (pH 7.2) [25,26].

The solutions for whole-cell patches were as follows. The pipette solution contained 132 mM KCl, 1 mM ethylene glycol-bis(β-aminoethylether)-*N*,*N*,*N*′,*N*′-tetraaceticacid (EGTA), 10 mM NaCl, 2 mM MgCl_2_, 10 mM HEPES, 1 mM Na_2_ATP, and 0.3 mM Na_2_GDP (pH = 7.2). The bath solution contained 142 mM NaCl, 2.8 mM KCl, 1 mM CaCl_2_, 1 mM MgCl_2_, 11 mM glucose, and 10 mM HEPES (pH = 7.4) [27,28,29].

CaCl_2_ was added to the pipette solutions to give a free Ca^2+^ ion concentration of 1.6 × 10^−6^ M in whole-cell experiments. The calculation of the free Ca^2+^ ion concentration in the pipette was performed using the MaxChelator software (Stanford University; Stanford, CA, USA). These intracellular Ca^2+^ ion concentrations are required for a stable seal formation in whole-cell experiments. 

### 2.4. Primary Murine Cell Culture

The bone and skeletal muscle tissues were collected from wild-type male and CS mice described above [9,12]. Briefly, the muscles were incubated with Dulbecco’s modified Eagle’s medium (DMEM+) solution (EuroClone S.p.A., Milano, Italy) composed of 1X antibiotics (1%), L-glutamine (1%), FBS (10%) (EuroClone, S.p.A., Milano, Italy) and enriched with collagenase type IX (0.1%) (SIGMA Chemical Co., Milan, Italy). The fibers were then enzymatically isolated from the Flexor digitorum brevis (FDB) mice muscles in the shaker incubator for 2 h [9,30,31].

The culture of primary bone cells from adult mice was obtained from long bone such as femora or calvaria, as previously described [9,31] Bones were collected, cleaned, and flushed to remove the internal bone marrow cells. Small bone pieces of around 1 mm^3^ were treated with trypsin-EDTA 0.25% (*w*/*v*) for 1 h and with 0.2% collagenase solution for an additional hour in a shaking water bath to remove all remaining soft tissue and adherent cells. Clean bone pieces have been cultured in a basal medium enriched with 50 µg/mL of ascorbic acid. As previously described, osteoblasts from calvaria chips were positive for Alizarin red staining and positive for PCR gene expression on cell pellets [32].

### 2.5. Patch-Clamp Experiments

Whole-cell patch-clamp experiments were performed in asymmetrical K^+^ ion concentration in physiological conditions using pipettes with a resistance of 3–5 MΩ. A drug solution was applied on the extracellular side. KATP currents of native skeletal muscle fibers were recorded in excised macropatches using pipettes of 0.9–1.2 MΩ of resistance on isolated skeletal muscle fibers in symmetrical 150 mM K^+^ ions concentrations on both sides of the membrane during pulse going from 0 mV to −60 mV (Vm) and drug solution directly applied on the internal side of the patches.

Drug actions on channel currents recorded during instantaneous whole-cell I/V relationships were investigated by applying a depolarization protocol in response to voltage pulses from −180 mV to +100 mV (Vm) in 20 mV steps. All experiments were performed at room temperature (20–22 °C) and sampled at 2 kHz (filter = 1 kHz) using an Axopatch-1D amplifier equipped with a CV-4 head stage (Axon Instruments, Foster City, CA, USA). Drug solutions were applied using a fast perfusion system during the continuous monitoring of the seal resistance at 18–22 °C room temperature in excised patches. In whole-cell patch-clamp experiments, before recording, cells were equilibrated for 10 s with the drug solution. The fast perfusion system is programmed at this time interval for seal stability. Data acquisition and analysis were performed using the pCLAMP 10 software suite (Axon Instruments, Foster City, CA, USA), as previously described. Seal resistance was continuously monitored during the experiment.

### 2.6. Statistical Analysis

The data are expressed as an average ± S.E.M. unless otherwise specified. The significance between data pairs was calculated by the paired Student’s *t*-test for *p* < 0.05. One-way ANOVA was used to evaluate the significance within and between data with a variance-ratio F > 1 at significance levels of *p* < 0.05.

The percentage of KATP current inhibition induced by ZOL was calculated as (I CTRL-I drug)/(I CTRL-I BaCl_2_) × −100. BaCl_2_ solution at high millimolar concentrations fully blocks all types of Kirs channels, including the Kir6.1 and Kir6.2 channels. The concentration–response data of ZOL against different KATP channel currents were fitted by the equation: ligand binding, four-parameter logistic function (linear) of Sigma Plot, 10.0.

## 3. Results

Previous studies showed that zoledronic acid (10^−12^–10^−4^ M) inhibited the KATP currents recorded in excised macropatches from isolated wild-type Extensor digitorum longus (EDL) and Soleus skeletal muscle fibers [31]. As shown in Figure 1, zoledronic acid also potently inhibited KATP channels in Flexor digitorum brevis (FDB) fibers from Kir6.1^wt/V65M^ mice, with an IC_50_ of ~1 μM and slope of ~0.6.

Intriguingly, the KATP currents of the heterozygous SUR2^wt/A478V^ muscles were less sensitive to zoledronic acid inhibition at a concentration above the IC_50_ but much lower slope (~0.3) but more sensitive at the lower range of concentrations, zoledronic acid failed to fully inhibit KATP currents in homozygous SUR2^A478V/A478V^ muscle, even at the highest concentrations (Figure 1). These results are in striking contrast to the effects of the sulfonylurea glibenclamide, which shows similar action on FDB fibers from WT and SUR2^wt/A478V^ but was less active on Kir6.1^wt/V65M^ as also seen in recombinant and native vascular smooth muscle channels [8,33,34,35] (Figure 2). The ability of glibenclamide to block KATP channels in Kir6.1^wt/V65M^ FDB muscle is in contrast to the lack of inhibitory effect in the soleus muscle [9,12,32,36,37] (Table 1). The KATP channels of SUR2^A478V/A478V^ were also less responsive to glibenclamide action.

In primary cultured bone calvaria cells, zoledronic acid was a very potent blocker of both WT and Kir6.1^wt/V65M^ KATP channels in whole-cell membrane patches when applied to the outside of the cell membrane (Figure 3). With an apparent IC_50_ of ~100 pM, this effect is considerably more potent than glibenclamide, which also inhibits both WT and mutant channels with IC_50_ of ~10 nM (Figure 3).

## 4. Discussion

Taken together with our previously published work [23], these data indicate that zoledronic acid inhibits KATP channels in the musculoskeletal system with varying potency in different cell types, where potency in femoral osteoblasts ≈ calvaria osteoblasts ≈ soleus muscle > EDL muscle ≈ FDB muscle (Table 2). Our previous gene expression analysis showed high expression of Kir6.1 and SUR2 in osteoblasts and soleus muscle [12,23,38] and we previously showed that ZOL inhibits recombinant KATP channels composed of Kir6.1/SUR2B subunits with increased potency relative to Kir6.2/SUR2B or Kir6.2/SUR1 channels [9]. Therefore, we propose that increased ZOL sensitivity in osteoblasts and soleus muscle likely reflects increased Kir6.1/SUR2B contributions to native KATP channels in these tissues relative to FBD and EDL muscle. Importantly, this is also consistent with our previous finding that the Kir6.1^V65M^ mutation, which reduces glibenclamide inhibition [11], has a greater effect on GLIB sensitivity of soleus KATP channels versus FDB KATP channels [9]. The high expression ratio of *KCNJ8*-Kir6.1/*KCNJ11*-Kir6.2 relates to the potency of the zoledronic acid in the musculoskeletal tissues (Table 2).

In contrast to reduced GLIB inhibition in native cells from Kir6.1^wt/V65M^ mice, here we show that the Kir6.1^V65M^ mutation does not significantly affect ZOL inhibition of KATP in either osteoblasts or FBD muscle. This suggests that the Kir6.1^V65M^ mutation does not markedly reduce ZOL effects and points to different molecular mechanisms of action for ZOL and GLIB inhibition. This finding might have potential implications for the treatment of CS patients: glibenclamide reverses CS-associated cardiovascular pathologies in SUR2^wt/A478V^ mutant mice but has reduced efficacy in Kir6.1^wt/V65M^ mice, leading to the suggestion that clinical efficacy of glibenclamide may be limited in Kir6.1-variant CS patients [10,39,40]. Our data here suggest that if ZOL is safely tolerated, or if other safe drugs can be identified that work via a similar mechanism, they might be more effective than sulfonylureas in Kir6.1-variant patients.

Curiously, we observed a marked reduction of ZOL inhibition in muscle from SUR2^A478V^ mutant mice. The mechanism for this effect is unknown and requires further study, but it might include direct effects on the binding site or allosteric effects of the mutation. In this respect, the KATP channels of the heterozygous SUR2^wt/A478V^ fibers showed a much lower slope than the channel in the WT/WT mice fibers. This can be related to the high-affinity interaction of the drug on the high-affinity sites on the channel complexes and the loss of interaction with lower-affinity sites, as supported by the fact that the channels were less sensitive to zoledronic acid inhibition at a concentration above the IC_50_. Lower affinity sites of zoledronic acid are either the nucleotide sites and sulfonylureas sites on Kir6.2-SUR1 and Kir6.1-SUR1 complexes as observed by docking analysis suggesting that conformational changes of these sites are responsible for the reduced sensitivity of the channels to zoledronic acid in the heterozygous SUR2^wt/A478V^.

In the present work, we demonstrate that zoledronic acid effectively and potently inhibits KATP currents in different skeletal muscle types and bone cells of Kir6.1^wt/V65M^ CS mice but not in the skeletal muscle of SUR2^A478V^ mice. It is a particularly potent inhibitor of KATP currents in soleus fibers and osteoblast cells. The effects of zoledronic acid in osteoblast and soleus muscle fibers of CS mice can be explained by the high expression/activity of the target channel subunits in these tissues vs. the fast-twitching fibers. Gene expression analysis indeed showed that osteoblasts and soleus fibers from WT/WT mice and rats express high Kir6.1-SUR2 channel proteins supporting this hypothesis [11,41,42,43].

One important point to consider is the very different sensitivities to zoledronic acid that are observed in skeletal muscle and osteoblasts. In WT EDL and FDB skeletal muscle, KATP channel inhibition IC_50_ is ~1 μM, when applied directly to the cytoplasmic face of the membrane in excised patch experiments. Conversely, in bone calvaria, the drug is ~3 orders of magnitude more potent, with IC_50_ ~1 nM, when applied outside the membrane. We suggest that zoledronic acid likely interacts with KATP channels on a cytoplasmic site. High uptake activity into osteoblast may raise the sub-membrane concentration substantially beyond bath concentrations in whole-cell studies. In some cells that did not belong to the musculoskeletal apparatus, the action of the drug applied on the extracellular side can be limited by the influx rate.

Previously, our docking simulation analysis [23,24] identified several potential binding sites for zoledronic acid on SUR2 subunits in the sulfonylurea and nucleotide binding regions and an additional site in the ATP binding region of Kir6 subunits. We identified in silico zoledronic acid binding at a site of SUR2 common to both SUR2A and SUR2B splice variants, composed of S401, T400, T687, W681, and S713 residues, at which zoledronic acid bound with binding energy and RMSD values comparable to those for ATP binding. Residue A475 (equivalent to the human A478 numbering) is located in a nearby region. Previous recombinant channel studies indicate that the SUR2^A478V^ mutation increases channel activity by promoting activation of the channel by MgADP binding to the SUR2 nucleotide binding folds (NBFs) [33,35], whereas the Kir6.1^V65M^ mutation acts to increase the intrinsic open state stability of the channel itself [5]. Most evidence points to Kir6.2/SUR2 being the primary subunit composition for KATP channels in skeletal muscle [14,30,39]. It also suggests that the primary action of zoledronic acid may be on the Kir6 subunit rather than the SUR2 subunit since the GoF is preserved in AV skeletal muscle channels, i.e., zoledronic acid inhibition is less effective when there is SUR2-dependent activation.

## 5. Conclusions

In conclusion, the inhibitory effects of zoledronic acid on KATP currents in Kir6.1^wt/VM^ CS mice observed in different skeletal muscle phenotypes, and bone cells but not in skeletal muscle of SUR2^A478V^ mice can help improve drug prescription in CS patients to treat musculoskeletal symptoms. Zoledronic acid is in the WHO’s List of Essential Medicines used as a primary therapy for skeletal disorders caused by imbalanced bone homeostasis [38], where osteoblast and osteoclast activities are not perfectly coupled, leading to excessive bone resorption like osteoporosis and Paget’s disease and Multiple Myeloma [40,41,42,43,44]. Therefore, being an antiproliferative and anticancer drug [45], it can be prescribed in those CS patients also affected by cancer. It can be a case for a novel drug repurposing in rare diseases in specific CS patients with Kir6.1^wt/VM^ mutations. It should be of note that zoledronic acid is a medication approved by both the FDA and EMA for bone diseases and muscular dystrophy in pediatric patients. It was shown that 24 months of e.v. zoledronic acid treatment, in addition to vitamin D and calcium, leads to improvements in bone mass scores compared with vitamin D and calcium alone in Duchenne Muscular Dystrophy patients treated with corticosteroids. It was generally found to be safe and well tolerated apart from marked acute-phase reaction on first exposure in these people with some benefit in reducing pain [45,46,47]. Similarly, it is effective in glucocorticoid-induced osteoporosis in chronic pediatric illnesses associated with osteoporotic fractures [46], but the mechanism of action is not known. In our opinion, it is likely that the reported case reports of rhabdomyolysis and myoglobinuria following the administration of zoledronic acid in Duchenne Muscular Dystrophy can be associated with the interactions with muscle KATP channels [47,48].

## Figures and Tables

**Figure 1 cells-12-00928-f001:**
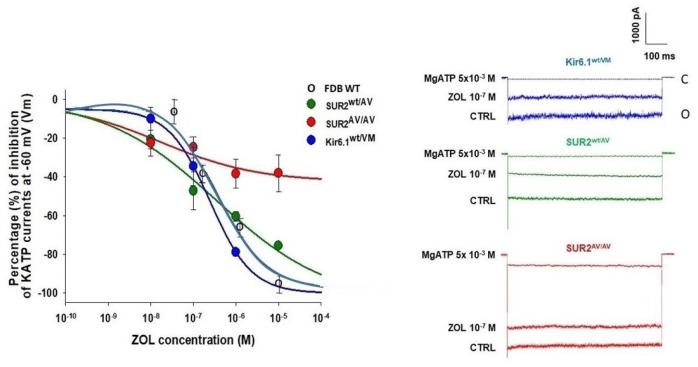
Concentration–response relationships of zoledronic acid concentrations vs. KATP current percentage inhibitions in the Flexor digitorum brevis (FDB) skeletal muscle fibers from different SUR2 hetero- and homozygous CS mice and relative KATP channel current traces. C—closed current level, O—open inward current level. The experiments were performed in excised macropatches from isolated fibers in the presence of high KCl on both sides of the membrane patches during voltage steps going from 0 mV to −60 mV (Vm). The drug solution was applied on the intracellular side of the patches. Zoledronic acid fully reduced KATP channel currents in either heterozygous SUR2^wt/AV^ and Kir6.1^wt/V65M^ but not in homozygous SUR2^A478V/A478V^ muscle fibers. Each point of the concentration–response curves was obtained on 11–19 patches.

**Figure 2 cells-12-00928-f002:**
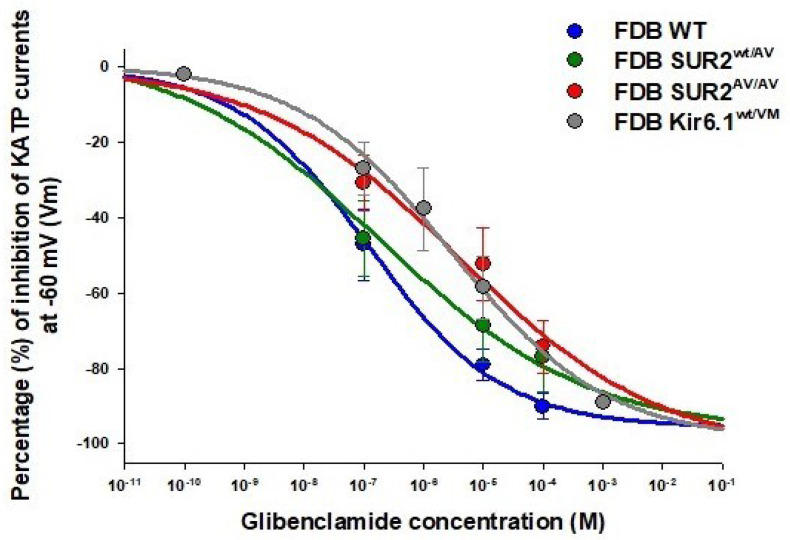
Concentrations-response relationships of glibenclamide concentrations vs. KATP current percentage inhibitions in FDB skeletal muscle fibers from SUR2^A478V^ homo and heterozygous and Kir6.1^wt/VM^ CS mice. The experiments were performed in excised macropatches from isolated fibers in the presence of high KCl on both sides of the membrane patches during voltage steps from 0 mV to −60 mV (Vm). The drug solution was applied on the intracellular side of the patches. The concentration–response curves of glibenclamide to reduce the KATP channel currents were significantly shifted to the right on the log concentration axis concerning the controls in homozygous SUR2^AV/AV^ and heterozygous Kir6.1^wt/VM^ CS mice. Each point of the concentration–response curves was obtained on 12–28 patches.

**Figure 3 cells-12-00928-f003:**
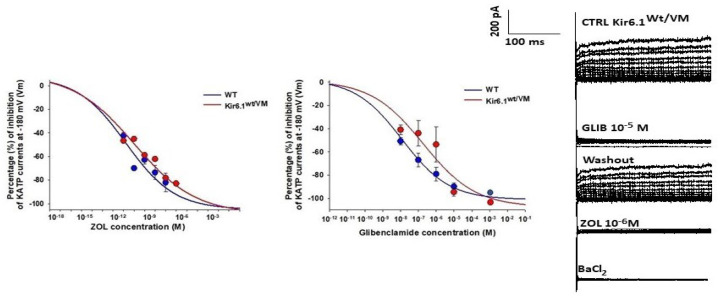
Concentration–response relationships of zoledronic acid (ZOL) and glibenclamide (GLIB) concentrations vs. KATP current percentage inhibitions in bone calvaria cells from heterozygous Kir6.1^wt/V56M^ CS mice. The experiments were performed in a whole-cell configuration in the physiological conditions of the membrane patches during voltage steps going from 180 mV to +100 mV (Vm). The drug solution was applied on the extracellular side of the patches using a fast perfusion system. The cells were equilibrated for 10 s with the drug solution before recordings. The concentration–response curves of glibenclamide to reduce the KATP channel currents of Kir6.1^wt/V56M^ osteoblasts were significantly shifted to the right on the log concentration axis concerning the WT/WT bone cells. Zoledronic acid potently inhibited the KATP currents in the Kir6.1^wt/V56M^ osteoblasts. Representative traces of KATP currents in osteoblasts related to the effects of glibenclamide and zoledronic acid on whole-cell KATP currents in osteoblasts are reported. The current is identified by the glibenclamide response applied to the patches. A washing period of 10 s is applied before recording. Each point of the concentration–response curves was obtained on 16–23 patches.

**Table 1 cells-12-00928-t001:** Fitting parameters of the concentrations-response relationships in different Cantú syndrome (CS) mice genotypes and cells.

Drugs	Mice Cell Types	IC_50_ (M)	Emax %	Slope	N Patches
Zoledronic acid	WT/WT fibers	8.02 ± 0.07 × 10^−7^	−99.5 ± 0.8	0.59 ± 0.04	11 (I-O)
	Kir6.1^wt/V65M^	* 1.1 ± 0.06 × 10^−7^	−100.3 ± 0.5	0.6 ± 0.02	18 (I-O)
	fibers				
	SUR2^wt/A478V^	3.5 ± 1.8 × 10^−7^	−104.7 ± 5.4	* 0.3 ± 0.06	19 (I-O)
	fibers				
	SUR2^A478V/A478V^	NA	−40.1 ± 2.4 *	* 0.02 ± 0.02	14 (I-O)
	fibers				
Glibenclamide	WT/WT	1.3 ± 0.2 × 10^−7^	−97.9 ± 1.3	0.6 ± 0.01	21 (I-O)
	fibers				
	Kir6.1^wt/V65M^	* 1.2 ± 0.1 × 10^−6^	−93.7 ± 1	0.4 ± 0.01	28 (I-O)
	fibers				
	SUR2^wt/A478V^	1.9 ± 2.2 × 10^−7^	−97 ± 7.5	0.25 ± 0.08	12 (I-O)
	fibers				
	SUR2^A478V/A478V^	* 4.2 ± 3.7 × 10^−6^	−102.5 ± 9	0.26 ± 0.07	18 (I-O)
	fibers				
Zoledronic acid	WT/WT	1.61 ± 2.2 × 10^−10^	−104.8 ± 8.7	0.2 ± 0.07	16 (W-C)
	osteoblasts				
	Kir6.1^wt/V65M^	1.1 ± 1.7 × 10^−10^	−108.6 ± 10.5	0.18 ± 0.05	20 (W-C)
	osteoblasts				
Glibenclamide	WT/WT	** 1.2 ± 0.8 × 10^−8^	−101.1 ± 3.5	0.3 ± 0.07	18 (W-C)
	osteoblasts				
	Kir6.1^wt/V65M^	** 1.8 ± 2 × 10^−7^	−107.5 ± 15	0.3 ± 0.2	23 (W-C)
	osteoblasts				

Fitting parameters of the concentration–response relationships of the currents recorded at −60 mV (Vm) in inside-out (I-O) and whole cell (W-C) patches vs. drug concentrations were calculated by using the equation: Ligand Binding; four-parameter logistic function (linear) of Sigma Plot, 10.0. Data significantly differ between * fiber data and ** osteoblasts data with one-way ANOVA for *p* < 0.05 and F > 1.

**Table 2 cells-12-00928-t002:** Potency summary of zoledronic acid against the KATP channel currents in different cells and relative gene expression data.

Drugs	Mice Cell Types	IC_50_ (M)	Genes/ Proteins	Gene Expr./Beta-Actin
Zoledronic acid	WT/WT Calvaria osteoblast	1.61 ± 2.2 × 10^−10^	*KCNJ8*/ Kir6.1	4.2123
			*KCNJ11*/ Kir6.2	1.1135
			*ABCC8*/ SUR1	1.34501
			*ABCC9*/ SUR2	0.99994
Zoledronic acid	Femoral osteoblasts	1.6 ± 2.8 × 10^−10^	*KCNJ8*/ Kir6.1	5.1121
			*KCNJ11*/ Kir6.2	1.3145
			*ABCC8*/ SUR1	1.56501
			*ABCC9*/ SUR2	1.19401
Zoledronic acid	Soleus fibers	2.1 ± 3.7 × 10^−10^	*KCNJ8*/ Kir6.1	1.9423
			*KCNJ11*/ Kir6.2	2.1315
			*ABCC8*/ SUR1	1.46751
			*ABCC9*/ SUR2	2.3294
Zoledronic acid	Flexor digitorum brevis fibers	8.0 ± 0.07 × 10^−7^	*KCNJ8*/ Kir6.1	1.2323
			*KCNJ11*/ Kir6.2	5.6185
			*ABCC8*/ SUR1	2.36751
			*ABCC9*/ SUR2	7.2294
Zoledronic acid	Extensor digitorum longus fibers	1.2 ± 1.4 × 10^−6^	*KCNJ8*/ Kir6.1	1.8423
			*KCNJ11*/ Kir6.2	6.13345
			*ABCC8*/ SUR1	1.6783
			*ABCC9*/ SUR2	8.6534

## Data Availability

The data are available at the corresponding author Domenico Tricarico.

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
