# Peer review of "Zoledronic Acid Blocks Overactive Kir6.1/SUR2-Dependent KATP Channels in Skeletal Muscle and Osteoblasts in a Murine Model of Cantú Syndrome"

_cells, 2023, doi:10.3390/cells12060928_

Round 1

Reviewer 1 Report

The manuscript from Scala et al. investigated the ability of zoledronic acid, to inhibit Kir6.1/SUR2 subtype KATP channels.  The authors use bone and skeletal muscle primary cultures from wild type (WT), and SUR2A478V (SUR2AV) and Kir6.2V65M (Kir6.1VM) mutant mice.  Using patch clamp elecytrophysiology, the authors demonstrated zolendronic acid reduced KATP channel currents in either heterozygous SUR2AV and Kir6.1(VM) cells from the flexor digiroum brevis (FDB) skeletal muscle fibers.  However zolendronic acid did not reduce KATP channel currents in homozygous SUR2AV mouse FDB muscle fibers.  Glibenclamide, a commonly used antagonist of KATP channels, did not show a change in inhibition across genotypes in FDB cells.  In primary bone cells, zolendronic acid was a blocker of WT and and Kir6.1VM mouse cells.  In the discussion, the authors conclude that, zolendronic acid inhibits KATP channels in the muscloskeletal system with varying potency (in?) different cell types, where potency is highest in bone cells followed by extensor digitorum longus (EDL) and soleus (SOL) muscles, and lastly in FDB muscles.  This is an interesting theory, but the authors do not show data to support their conclusion that, “increases ZOL sensitivity in osteoblasts and SOL muscle likely reflects increased Kir6.1/SUR2B contributions to native KATP channels in these tissues relative to FBD and EDL muscle.”  If differences in KATP channel subtype expression are known, perhaps it would make sense to plot IC50 values versus mRNA/protein expression to determine a quantitative correlation.  Furthermore, it is difficult to determine if osteoblasts have a difference in sensitivity due to the lack of data from the SUR2AV mice to compare to the data collected from the FDB mice.  

  • The percentage of KATP current inhibition induced by zolendronic acid was calculated by using barium chloride as a control.  Listing a rationale for this might help readers not familiar with KATP channel function.

  • A table indicating the drug treatment, genotype, and cell type along with IC50 and Hill Slope would help the reader to make comparisons across all treatment conditions.  

  • Consistent nomenclature (e.g. SUR2A478V vs SUR2AV) should be used throughout the manuscript

  • The methods section mentions dissection of both extensor digitorum longus (EDL) and soleus (SOL) muscles - no data is shown using these cell types

  • In the discussion section, the authors state, “Kir6.2/SUR2 being the primary subunit composition for KATP channels in skeletal muscle. This may explain why the zolendronic acid effectively inhibits skeletal muscle KATP channel sin VM animals, but less so in AV animals.”  The authors pointed out the other types of skeletal muscle do not show this pattern of inhibitory differences across genotypes, so this statement does not seem to make sense.

  • Zolendronic acid is a medication approved by both the FDA and EMA for bone diseases and muscular dystrophy.  Does this manuscript indicate that loss of KATP channel activity contributes to this drug’s use in any of the approved indications/diseases?

Author Response

  • The percentage of KATP current inhibition induced by zoledronic acid was calculated by using barium chloride as a control.  Listing a rationale for this might help readers not familiar with the KATP channel function.
  • Ok
  • A table indicating the drug treatment, genotype, and cell type along with IC50 and Hill Slope would help the reader to make comparisons across all treatment conditions. 
  • OK, table 1 has been introduced as requested
  • Consistent nomenclature (e.g. SUR2A478V vs SUR2AV) should be used throughout the manuscript
  • OK
  • The methods section mentions the dissection of both extensor digitorum longus (EDL) and soleus (SOL) muscles - no data is shown using these cell types
  • OK
  • In the discussion section, the authors state, “Kir6.2/SUR2 being the primary subunit composition for KATP channels in skeletal muscle. This may explain why the zoledronic acid effectively inhibits skeletal muscle KATP channel in VM animals but less in AV animals.”  The authors pointed out that the other types of skeletal muscle do not show this pattern of inhibitory differences across genotypes, so this statement does not seem to make sense.
  • The reviewer is right and the sentence is deleted
  • Zoledronic acid is a medication approved by both the FDA and EMA for bone diseases and muscular dystrophy.  Does this manuscript indicate that loss of KATP channel activity contributes to this drug’s use in any of the approved indications/diseases?
  • The reviewer's idea is intriguing, zoledronic acid is a medication approved by both the FDA and EMA for bone diseases and muscular dystrophy in pediatric patients. It was shown that 24 months of IV zoledronic acid treatment in addition to vitamin D and calcium leads to improvements in bone mass scores compared with vitamin D and calcium alone in DMD patients treated with corticosteroids. It was generally found to be safe and well tolerated apart from a marked acute-phase reaction on first exposure in these people with some potential benefit in reducing pain (Zacharin et al., 2021). Similarly, it is effective in glucocorticoid-induced osteoporosis in chronic pediatric illnesses that are associated with osteoporotic fractures (Ward et al., 2021). While in CS the gain of function mutations of the KCNJ11/9 and ABCC8/9 genes are associated with abnormal bone development and the use of zoledronic acid may help in inhibiting the overactive Kir6.2V65M mutant, but we do not have evidence about the KATP channels response to zoledronic acid on other conditions like DMD or other bone diseases. These notes were introduced in the conclusion sections and the relative references.

Randomized Controlled Trial Evaluating the Use of Zoledronic Acid in Duchenne Muscular Dystrophy.

Zacharin M, Lim A, Gryllakis J, Siafarikas A, Jefferies C, Briody J, Heather N, Pitkin J, Emmanuel J, Lee KJ, Wang X, Simm PJ, Munns CF.J Clin Endocrinol Metab. 2021 Jul 13;106(8):2328-2342. doi: 10.1210/clinem/dgab302.

Zoledronic Acid vs Placebo in Pediatric Glucocorticoid-induced Osteoporosis: A Randomized, Double-blind, Phase 3 Trial.

Ward LM, Choudhury A, Alos N, Cabral DA, Rodd C, Sbrocchi AM, Taback S, Padidela R, Shaw NJ, Hosszu E, Kostik M, Alexeeva E, Thandrayen K, Shenouda N, Jaremko JL, Sunkara G, Sayyed S, Aftring RP, Munns CF.J Clin Endocrinol Metab. 2021 Nov 19;106(12):e5222-e5235. doi: 10.1210/clinem/dgab458.

On the other hand, case reports of rhabdomyolysis and myoglobinuria (Ivanyuk et al., 2018; Lemon et al., 2019) have been reported that in our opinion can be associated with interaction with muscle KATP channels (Maqoud et al., 2021). Also, these considerations and the relative references were included in the conclusion section.

Rhabdomyolysis and myoglobinuria following bisphosphonate infusion in patients with Duchenne muscular dystrophy.

Lemon J, Turner L, Dharmaraj P, Spinty S.Neuromuscul Disord. 2019 Jul;29(7):567-568. doi: 10.1016/j.nmd.2019.05.002. Epub 2019 May 8.

Myoglobinuria in two patients with Duchenne muscular dystrophy after treatment with zoledronate: a case report and call for caution.

Ivanyuk A, García Segarra N, Buclin T, Klein A, Jacquier D, Newman CJ, Bloetzer C.Neuromuscul Disord. 2018 Oct;28(10):865-867. doi: 10.1016/j.nmd.2018.08.004. Epub 2018 Aug 10.

Maqoud et al., “Zoledronic Acid as a Novel Dual Blocker of KIR6 . 1 / 2-SUR2 Subunits of ATP-Sensitive K + Channels : Role in the Adverse Drug Reactions,” pp. 1–17, 2021.

Reviewer 2 Report

The manuscript entitled “Zoledronic Acid Blocks Overactive Kir6.1/SUR2-Dependent KATP Channels in Skeletal Muscle and Osteoblasts of a Murine Model of Cantú Syndrome” by Scala et al. described KATP current responses as a function of pharmacological drug concentration in Cantu syndrome(CS) mouse models. Specifically, the authors investigated the action of zoledronic acid and glibenclamide, targeting KATP channels of flexor digitorum brevis (FDB) skeletal muscle fibers and bone calvaria cells of Kir6.1V65M and SUR2A478V CS mice to evaluate the possible use in the treatment of the musculoskeletal symptoms in CS. The paper is very descriptive, yet the findings are interesting to publish, but the manuscript needs major improvement for experimental clarity and readability.

Abstract.

Patch-clamp experiments showed that zoledronic acid (ZOL) fully reduced the KATP currents recorded in cell-attached patches in bone calvaria cells from WT/WT and heterozygous Kir6.1wt/V65MCS mice, with IC50 for ZOL block<1nM in each case.” This finding is from Figure 3. Would it be better to move to the end of Figures 1 and 2 findings in the Abstract?

In many places, references are listed, like Huang et al., 2018 (line 88), line 240, line 244 … Please add the correct reference numbers.

Line 177, IC50 of ~ 1mM. Please check the unit.

Line 226, -1.1 ± 1.7 x 10-10. Add unit and delete -, if necessary

Figure 3. The authors must clarify how these data were obtained. The provided description in the Method section is misleading (line 116): “The solutions for whole-cell and cell attached patches were as follow…”. The composition of the pipette solution mimics the intracellular milieu including ATP and GDP, that would be suitable for the whole-cell patch clamp recording. However, the Figure 3 legend indicates the cell-attached patch usage (line 220). What is the reason for loading pipettes with the intracellular solution under such recording conditions? If it is the cell-attached patch, what is the actual membrane patch potential (Epatch=Ecell – Epipette)? How under the cell-attached condition the KATP current was induced? With indicated pipette resistance (line 146) under cell-attached mode, the authors must monitor a single channel activity. How was the KATP channel activity identified? What is the timing of the drug application during the recordings? The representative traces must be provided.

Of note, despite mentioning whole-cell conditions in the Methods, the reviewer is unable to find any data obtained using such patch-clamp technique throughout the manuscript. If the author made a mistake and implemented the whole-cell technique for Fig.3, all sets of raised questions remain actual that the authors must address.

In addition, they must indicate which formula of the HilI equation they used to construct the dose-response curves. It looks strange that at the lowest drug concentrations, the curve is above 0 value (Fig.3, left panel), which contradicts the classical law expression.

Please add dose-response data point at mM glibenclamide concentration (Fig. 3. right panel) . With this addition, the fitting will be much reliable.

Discussion

Line 235, is -> in

Line 233, “Taken together with our previously published work [24], these data indicate that zoledronic acid inhibits KATP channels in the musculoskeletal system with varying potency is different cell types, where potency in femoral osteoblasts ≈ calvaria osteoblasts ≈ soleus muscle > EDL muscle ≈ FDB muscle.” In connection with this sentence, please add a potency summary Table, including femoral osteoblasts, calvaria osteoblasts, soleus muscle, EDL muscle, and FDB muscle, to enhance the readability. In addition, as the authors published gene expression analysis of Kir6.1 and SUR2 in osteoblast and soleus muscle, it would be helpful to include this information in the table, too.

Line 270, One important point to consider is the very different sensitivities to the drug….. Can you specify the drug name?

Line 276, [ref], please add a reference number

Line 295, please use a new line to start with 5. Conclusions

Reference numbers 23 and 24 indicate the same paper. Please correct.

Author Response

Abstract.

Patch-clamp experiments showed that zoledronic acid (ZOL) fully reduced the KATP currents recorded in cell-attached patches in bone calvaria cells from WT/WT and heterozygous Kir6.1wt/V65MCS mice, with IC50 for ZOL block<1nM in each case.” This finding is from Figure 3. Would it be better to move to the end of Figures 1 and 2 findings in the Abstract?

OK

In many places, references are listed, like Huang et al., 2018 (line 88), line 240, line 244 … Please add the correct reference numbers.

OK

Line 177, IC50 of ~ 1mM. Please check the unit.

OK

Line 226, -1.1 ± 1.7 x 10-10. Add unit and delete -, if necessary

OK

Figure 3. The authors must clarify how these data were obtained. The provided description in the Method section is misleading (line 116): “The solutions for whole-cell and cell attached patches were as follow…”. The composition of the pipette solution mimics the intracellular milieu including ATP and GDP, that would be suitable for the whole-cell patch clamp recording. However, the Figure 3 legend indicates the cell-attached patch usage (line 220). What is the reason for loading pipettes with the intracellular solution under such recording conditions? If it is the cell-attached patch, what is the actual membrane patch potential (Epatch=Ecell – Epipette)? How under the cell-attached condition the KATP current was induced? With indicated pipette resistance (line 146) under cell-attached mode, the authors must monitor a single channel activity. How was the KATP channel activity identified? What is the timing of the drug application during the recordings? The representative traces must be provided.

Ok representative traces related to the effects of glibenclamide and zoledronic acid on whole cell KATP currents not cell-attached patches in osteoblasts have been added to figure 3 as requested. The current is identified by the response to glibenclamide.

Of note, despite mentioning whole-cell conditions in the Methods, the reviewer is unable to find any data obtained using such patch-clamp technique throughout the manuscript. If the author made a mistake and implemented the whole-cell technique for Fig.3, all sets of raised questions remain actual that the authors must address.

The reviewer is absolutely right it was a mistake the data were obtained using whole-cell recording, not cell-attached patches that were used in some preliminary experiments, but all pharmacology data on osteoblasts was performed in Whole cell mode. We apologize for this and we thank the reviewer for her/his important note, therefore to avoid confusion we revised the methods and figure 3 legends accordingly.

In addition, they must indicate which formula of the HilI equation they used to construct the dose-response curves. It looks strange that at the lowest drug concentrations, the curve is above 0 value (Fig.3, left panel), which contradicts the classical law expression.

The reviewer is right, the Equation: Ligand Binding; four-parameter logistic function (linear) of Sigma Plot, 10.0 was used.

Please add dose-response data point at mM glibenclamide concentration (Fig. 3. right panel). With this addition, the fitting will be much more reliable.

OK

Discussion

Line 235, is -> in

OK

Line 233, “Taken together with our previously published work [24], these data indicate that zoledronic acid inhibits KATP channels in the musculoskeletal system with varying potency is different cell types, where potency in femoral osteoblasts ≈ calvaria osteoblasts ≈ soleus muscle > EDL muscle ≈ FDB muscle.” In connection with this sentence, please add a potency summary Table, including femoral osteoblasts, calvaria osteoblasts, soleus muscle, EDL muscle, and FDB muscle, to enhance the readability. In addition, as the authors published gene expression analysis of Kir6.1 and SUR2 in osteoblast and soleus muscle, it would be helpful to include this information in the table, too.

We thank the reviewer for his/her nice idea and accordingly, we added a new summary table 2 in the discussion section

Line 270, One important point to consider is the very different sensitivities to the drug….. Can you specify the drug name?

OK

Line 276, [ref], please add a reference number

OK

Line 295, please use a new line to start with 5. Conclusions

OK

Reference numbers 23 and 24 indicate the same paper. Please correct.

OK

Reviewer 3 Report

This paper by Scala et al investigates the effects of zoledronic acid on KATP channels in bone cells and muscle fibres isolated from mice carrying gain-of-function human mutations in the subunits of the KATP channels (SUR2 and Kir6.1) implicated in Cantú syndrome. Using patch clamping the authors demonstrate differential effects of zoledronic acid in muscle and bone cells isolated from heterozygous Kir6.1wt/V65M mice, heterozygous SUR2wt/A478Vmice and homozygous SUR2A478V/A478V mice. It is suggested that the effectiveness of zoledronic acid as an inhibitor depends on the subunit composition of KATP channels. Presented data is potentially important for development of novel therapies for people with Cantú syndrome.

There are several points that require clarification:

11.     Line 122. It is stated that free Ca2+ ion concentration in whole-cell experiments was 1.6 × 10−6 M. This is 10 times higher than a resting cytosolic Ca2+ concentration. What was the reason for using high Ca2+ levels in the cells, and could this have affected the activity of BK channels during the recordings? 

22.  Line 147. “…drug solution was applied on the external side.”  It would be better to use terms “intracellular, extracellular” than “internal, external” in regards to the drug applications. Furthermore, this part of the methods needs further explanations. In the cell-attached experiments, was the drug applied in the patch pipette, in the bath, or both?  

33. Line 158. “In cell-attached patch-clamp experiments, before recording, the cell attached patches were equilibrated for 10 s with the drug solution”. I’m assuming that the drug was applied to the bath after establishing the seal and recording the control currents. What was the criteria of waiting for 10 s and not longer?

44. Figure 1. The dose-response curve for the WT data does not look right at the lowest concentration range. It seems that this portion of the curve was replaced with a straight line.

55. Line 191. “…the KATP currents of the heterozygous SUR2wt/A478V muscles was less sensitive to zoledronic acid inhibition…” This is not quite correct. It is less sensitive at concentrations above EC50, but more sensitive at the lower range of concentrations.

66.  The authors need to include some discussion about the mechanistic significance of the Hill coefficient changes

77. The references 23 and 24 duplicate each other.

Minor:

Line 64 “…with vacuolar future [16]…”   Feature?

Line 71 “There are very few reports are available...”

Line 276 “… into osteoblasts[ref]”

Author Response

There are several points that require clarification:

  1.    Line 122. It is stated that free Ca2+ ion concentration in whole-cell experiments was 1.6 × 10−6 M. This is 10 times higher than a resting cytosolic Ca2+ concentration. What was the reason for using high Ca2+ levels in the cells, and could this have affected the activity of BK channels during the recordings? 

The reason is related to the stability of the seal in the whole cell that is enhanced using this free Ca2+ ions concentration. This is now clarified in the methods. BK channel can be activated but the cells not responding to glibenclamide were excluded, we had 10 osteoblasts excluded from our experiments.

  1. Line 147. “…drug solution was applied on the external side.”  It would be better to use the terms “intracellular, extracellular” than “internal, external” in regard to the drug applications. Furthermore, this part of the method needs further explanations. In the cell-attached experiments, was the drug applied in the patch pipette, in the bath, or both?  

OK

we clarify that the data are referred to whole cells in osteoblast and not to cell-attached patches and we apologize for the erroneous reports in the methods and results that were indeed revised.

  1. Line 158. “In cell-attached patch-clamp experiments, before recording, the cell-attached patches were equilibrated for 10 s with the drug solution”. I’m assuming that the drug was applied to the bath after establishing the seal and recording the control currents. What were the criteria for waiting for 10 s and not longer?

The same response as before, this time of preincubation of 10 s is applied before recording for seal stability and for allowing drug influx during drug solution exchange.

  1. Figure 1. The dose-response curve for the WT data does not look right at the lowest concentration range. It seems that this portion of the curve was replaced with a straight line.

The reviewer is right the WT data of figure 1 was refitted

  1. Line 191. “…the KATP currents of the heterozygous SUR2wt/A478V muscles was less sensitive to zoledronic acid inhibition…” This is not quite correct. It is less sensitive at concentrations above EC50, but more sensitive at the lower range of concentrations.

This consideration is correct again we thank the reviewer for this note, which was included in the results and in the discussion section

  1. The authors need to include some discussion about the mechanistic significance of the Hill coefficient changes

The same as above

  1. References 23 and 24 duplicate each other.

OK

Minor:

Line 64 “…with vacuolar future [16]…”   Feature?

Line 71 “There are very few reports are available...”

Line 276 “… into osteoblasts[ref]”

OK

Round 2

Reviewer 1 Report

Thank you for incorporating the reviewer's comments.

Reviewer 2 Report

Recommend for publication.